# IL11 Stimulates IL33 Expression and Proinflammatory Fibroblast Activation across Tissues

**DOI:** 10.3390/ijms23168900

**Published:** 2022-08-10

**Authors:** Anissa A. Widjaja, Sonia Chothani, Sivakumar Viswanathan, Joyce Wei Ting Goh, Wei-Wen Lim, Stuart A. Cook

**Affiliations:** 1Cardiovascular and Metabolic Disorders Program, Duke-National University of Singapore Medical School, Singapore 169857, Singapore; 2National Heart Research Institute Singapore, National Heart Centre Singapore, Singapore 169609, Singapore; 3MRC-London Institute of Medical Sciences, Hammersmith Hospital Campus, London SW7 2AZ, UK

**Keywords:** IL-33, IL-11, IL-6, fibrosis, fibroblasts, inflammation, interleukin, transcription, translation

## Abstract

Interleukin 11 (IL11) is upregulated in inflammatory conditions, where it is mostly believed to have anti-inflammatory activity. However, recent studies suggest instead that IL11 promotes inflammation by activating fibroblasts. Here, we assessed whether IL11 is pro- or anti-inflammatory in fibroblasts. Primary cultures of human kidney, lung or skin fibroblasts were stimulated with IL11 that resulted in the transient phosphorylation of signal transducer and activator of transcription 3 (STAT3) and the sustained activation of extracellular signal-regulated protein kinases (ERK). RNA sequencing over a time course of IL11 stimulation revealed a robust but short-lived transcriptional response that was enriched for gene set hallmarks of inflammation and characterized by the upregulation of *SERPINB2*, *TNFRSF18*, *Interleukin 33 (IL33)*, *CCL20, IL1RL1*, *CXCL3/5/8*, *ICAM1* and *IL11* itself. *IL33* was the most upregulated signaling factor (38-fold, *p* = 9.8 × 10^−5^), and *IL1RL1*, its cognate receptor, was similarly increased (18-fold, *p* = 1.1 × 10^−34^). In proteomic studies, IL11 triggered a proinflammatory secretome with the notable upregulation of IL8, IL6, MCP1, CCL20 and CXCL1/5/6, which are important chemotaxins for neutrophils, monocytes, and lymphocytes. IL11 induced IL33 expression across fibroblast types, and the inhibition of STAT3 but not of MEK/ERK prevented this. These data establish IL11 as pro-inflammatory with specific importance for priming the IL33 alarmin response in inflammatory fibroblasts across tissues.

## 1. Introduction

Interleukin 11 (IL11) is a little-studied member of the IL6 family of cytokines [1,2]. IL11 was initially characterized as anti-inflammatory, anti-fibrotic and pro-regenerative based largely on experiments in which recombinant human IL11 (rhIL11) was administered to mouse models of disease [3]. However, species-matched IL11 was recently found to have ERK-dependent pro-fibrotic activity in fibroblasts [4,5,6,7], and it transpires that rhIL11 has unexpected activity in the mouse and paradoxically inhibits endogenous murine IL11 [8]. In mammals, IL11 is now largely accepted to be pro-fibrotic [9] and is increasingly viewed as anti-regenerative [8,10,11].

The idea that IL11 may be anti-inflammatory remains pervasive: a detailed review of the role of IL11 in inflammatory diseases published in 2022 [12] concluded “whether this [IL11] elevation is pathogenic or a natural host response to restore homeostasis is not clear.” Indeed, multiple studies that used rhIL11 in mouse models showed anti-inflammatory effects, and based on this, IL11 was proposed as a therapeutic for inflammatory diseases [13,14,15,16]. The absolute conviction of IL11′s anti-inflammatory activity is apparent from clinical trials in which rhIL11 was administered to patients with hepatitis, colitis, rheumatoid arthritis and other conditions [12].

Newer data challenge the earlier literature on the role of IL11 in inflammation as its activity in fibroblasts is now thought to promote inflammation in colitis, synovitis and cancers [11,17,18,19]. Furthermore, the expression of IL11 in vascular smooth muscle cells is linked with inflammation [20,21]. Similarly, IL11 activity in hepatic with pancreatic stellate cells seems pro-inflammatory [22,23], and IL11 activity in cancer-associated fibroblasts is linked with an inflammatory response [18,24,25,26].

In fibroblasts, IL11 signals via IL11RA/IL6ST in a hexameric complex to activate both JAK/STAT3 and MEK/ERK, and possibly AKT [7]. MEK/ERK activity is associated with the pro-fibrotic effects of IL11, which are seen late (>8 h) after IL11 stimulation of fibroblasts in vitro and mediated at the level of translation [6]. The early (<1 h) IL11-induced phosphorylation of STAT3 in fibroblasts seems unrelated to fibrogenesis, at least in vitro [7], but the physiological consequences of its activation are not known. Here, we performed a series of signaling, RNA-sequencing, proteomic and pharmacologic experiments to determine whether IL11 is pro- or anti-inflammatory in fibroblasts.

## 2. Results

### 2.1. Time Course of IL11 Signaling in Fibroblasts from Three Tissues

We stimulated primary cultures of human fibroblasts from kidney (HKF), lung (HLF) and skin (HSF) with IL11 (10 ng/mL). Using immunoblotting, we assessed the activity of the major IL6ST (gp130) signaling modules (MEK/ERK, JAK/STAT3, AKT) over a 24 h time course (Figure 1).

Across fibroblast types, IL11 induced STAT3 phosphorylation (pSTAT3) at 5 min that remained elevated until 30 min and returned to basal levels by 1 h (Figure 1A–C). In HKF and HLF, pERK levels were sustained and bimodally increased, with the highest levels observed at 5 min and 24 h poststimulation (Figure 1A,B). In HSF, pERK levels notably increased from 5 to 30 min and then diminished, although they remained elevated compared with baseline at 24 h (Figure 1C). AKT phosphorylation (pAKT) was inconsistent with pAKT, being bimodally elevated in IL11-treated HKF but downregulated below baseline at 5 to 30 m following IL11 stimulation in HLF and HSF.

The activation of fibroblasts with IL11 causes ERK-dependent fibroblast-to-myofibroblast transformation and the expression of alpha-smooth muscle actin (ɑSMA), which occurs in the absence of detectable *ACTA2* transcript increase in vitro [6,7]. We probed for ɑSMA across the time courses and observed consistent and late (6–24 h) upregulation of this myofibroblast marker across fibroblast types (Figure 1A–C).

These data show in primary human fibroblasts from three different body sites that IL11 induces early and transient STAT3 activation, bimodal and sustained ERK activation and inconsistent activation or inhibition of AKT. IL11-stimulated ɑSMA occurred late in the time course, when ERK remains activated but STAT3 is not, in keeping with the ERK dependency of this translationally driven effect, which has been repeatedly observed [6,7,20].

### 2.2. IL11 Activates a Pro-Inflammatory Transcriptional Response in Fibroblasts

IL11 activates STAT3 in a range of cell types, including fibroblasts [12,18,27,28]. While the role of IL11 in STAT3 activation appears unrelated to fibrogenesis [7], we hypothesized that the early activation of STAT3 following IL11 exposure might lead to a STAT3-driven transcriptional response. To test this, we performed RNA sequencing of HKF stimulated with IL11 over a time course (1, 6 or 24 h).

In cardiac fibroblasts, we have shown previously that IL11 has little transcriptional effect 24 h after stimulation, when profibrotic gene translation is profound [6]. In agreement with this, principal component analysis (PCA) showed that the gene expression patterns of HKF 24 h post-IL11 stimulation were similar to unstimulated cells (Figure 2A). PCA also showed that the transcriptomes of HKF stimulated with IL11 for 1 or 6 h were notably different from baseline and from each other (Figure 2A). The inspection of individual gene expression showed large numbers of significantly differentially regulated genes in IL11-stimulated fibroblasts at 1 and 6 h post-stimulation but little effect at 24 h (Figure 2B and Appendix A).

Using the molecular signatures database (MSigDB), we assessed for hallmark gene set enrichment following IL11 stimulation (Appendix A). Across the time course (1, 6 and 24 h), IL11 significantly (*p* < 7 × 10^−5^) induced the hallmark gene sets “TNFA_signaling_via_NFKB” and “Epithelial_Mesenchymal_Transition (EMT)”. At early time points (1 and 6 h), there was a concordant and significant (*p* < 7 × 10^−5^) upregulation of the hallmark gene sets “Inflammatory_response”, “MYC_Targets”, “KRAS_Signaling_Up” and ‘Angiogenesis”, among others. The hallmark gene sets “IL6_JAK_STAT3_Signaling” and “IL2_STAT5_Signaling’’ were variably activated across the time course. On the single-gene level, there was a significant upregulation of a number of pro-inflammatory factors at 6 h, which included *SERPINB2*, *TNFRSF18*, *IL33*, *CCL20*, *IL1RL1*, *CXCL3/5/8*, *ICAM1*, *IFNE* and *IL11* itself (Figure 2B and Appendix A).

On RNA sequencing, the most-upregulated signaling factor after IL11 stimulation was *IL33* (37-fold, 9.83 × 10^−5^; 6 h poststimulation), which was also increased (13-fold) at 1 h, albeit non-significantly, and returned to basal levels of expression by 24 h (Figure 2B and Appendix A). *IL1RL1,* the cognate IL33 receptor that binds to *IL1RAP* as a heterodimer to activate NFκB and MAPKs, was also significantly upregulated (18-fold, 1.13 × 10^−34^; 6 h poststimulation) (Appendix A). We validated the RNAseq data for IL33 at 1 h and 6 h using QPCR and extended analyses to include both HLF and HSF, which also significantly upregulated *IL33* at these time points (Figure 2C,D).

We tested whether the IL11-induced upregulation of *IL33* was related to either STAT3 or ERK activity using specific inhibitors of either STAT3 (Stattic [29]) or MEK/ERK (U0126) [7]. Cells were incubated with inhibitors and IL11 for 1 h to limit inhibitor toxicities and/or secondary effects, and IL33 mRNA levels were determined with qPCR. This showed that IL11-dependent IL33 upregulation is STAT3-dependent but unrelated to MEK/ERK activity (Figure 2D).

### 2.3. Fibroblasts Stimulated with IL11 Secrete a Range of Pro-Inflammatory Factors

Our RNA sequencing studies revealed an early pro-inflammatory transcriptional response in fibroblasts stimulated with IL11. However, we, and others, have shown that IL11 has translation-specific effects that are apparent at later time points in vitro [5,6]. Therefore, we profiled the fibroblast secretome using the Olink inflammation proteomics panel on supernatants from HKF stimulated with IL11 over a 24 h time course (Figure 3).

We detected a range of pro-inflammatory factors that were increased by IL11 stimulation over the time course. There were notable increases in chemotactic factors for neutrophils (IL6, IL8, CXCL1/5/6), lymphocytes (CCL20) and also monocytes (MCP1) (Figure 3A,B). All these proinflammatory factors showed significant, time-dependent increases, with the highest levels 24 h after stimulation, a time point when the transcriptome had already returned to baseline (Figure 2A). In keeping with IL33 being an intracellular alarmin [30], there was only a small increase in IL33 detected in the supernatants (FC = 2.5, *p* < 0.001; 24 h). These data confirm and extend the finding that IL11 is proinflammatory (Figure 2C) and reinforce the notion of translation-specific activities of IL11 in fibroblasts.

### 2.4. IL11 Stimulates STAT3-Dependent IL33 Upregulation in Fibroblasts

To examine whether IL11-stimulated IL33 expression mirrors the transcriptional response or if there are translation-specific effects as well, we profiled IL33 by immunoblotting over a time course of IL11 stimulation in HKF, HSF and HLF (Figure 4A–C). This showed that IL33 protein levels largely mirror *IL33* transcript variation and a tight coupling of transcription and translation of IL33 in fibroblasts. We further confirmed that the inhibition of STAT3 with Stattic inhibited IL33 expression following IL11 stimulation (Figure 4D).

## 3. Discussion

There is a large body of older literature that teaches that IL11 is anti-inflammatory [12,31,32,33,34]. The power of this belief led to clinical trials of rhIL11 in patients with Crohn’s disease [35,36], hepatitis [37], rheumatoid arthritis [38] and psoriasis [39]. In 2022, a review of IL11 in inflammation concluded that its role may be homeostatic and anti-inflammatory [12]. However, many of the original insights into IL11 biology were based on the use of rhIL11 in the mouse, which has unexpected effects [7,8]. This realization underpins the discovery that IL11 is profibrotic [4,5,6,9,11], whereas the earlier literature had reported it as anti-fibrotic [3,40,41,42]. We suggest that, as it was for fibrosis, the role for IL11 in inflammation has been misinterpreted.

Here, we define the dynamic transcriptional landscape of fibroblasts following IL11 stimulation. The time course used is of particular importance as we, and others, have struggled to detect an effect of IL11 on transcription in stromal cells (e.g., fibroblasts, vascular smooth muscle cells or hepatic stellate cells) in vitro at late time points (24 h) after stimulation, when translational effects are abundant [5,6]. We documented robust and significant transcriptional effects of IL11 at 1 and 6 h post stimulation, when both STAT3 and ERK are phosphorylated across fibroblast types.

The hallmark gene sets enriched following IL11 stimulation are consistent with a pro-inflammatory effect of IL11 in fibroblasts. The individual genes upregulated included chemotaxins (e.g., *CCL20*, *CXCL3/5/8*), *IL33* and *IL11* itself. These data identify IL11 as pro-inflammatory in fibroblasts and reaffirm the autocrine loop of IL11-induced *IL11*.

IL33 is an alarmin important for immunity, inflammation, epithelial barrier function and cancer [30,43], and it was the most upregulated signaling factor following IL11 exposure. IL33′s cognate receptor *IL1RL1* (ST2) also increased. Of note, IL11-expressing colonic fibroblasts have high levels of *Il1rl1*, along with *Il13ra2* and *Cxcl5* [19,24]. We found that IL11-induced IL33 upregulation is STAT3-dependent and unrelated to ERK activity. In keeping with this, IL11 induces IL33 in tumors via a Stat3 response element in the 5′-region of the mouse *Il33* gene and an IL11/IL33 interaction is described in colonic polyposis [43,44]. Thus, IL11-stimulated fibroblasts are primed to release IL33 when damaged to activate IL33 pathways in neighboring IL1RL1-expressing fibroblasts and other cells, which include immune cells in cancers [45].

Our proteomic studies show that IL11 induces a pro-inflammatory secretome from fibroblasts. Secreted proteins include genes identified by RNA-seq but with additional factors apparent at later time points, when the transcriptome has returned to baseline. Of the secreted factors, many are chemotaxins for neutrophils, lymphocytes and monocytes. It was notable that the IL11-induced secretome comprises key factors of the senescence associated secretory phenotype (SASP), including IL6, IL8 and CCL20, which are also important in the tumor microenvironment [25,46].

IL11 evolved in the fish, where it is constitutively expressed in the gills, which are exposed to pathogens [47]. In the fish, IL11 is strongly upregulated across tissues in response to viral, bacterial or parasitic infection [48,49,50]. In the pig, IL11 is one of the most upregulated genes in the colon, along with type 2 immune genes, following *Trichuris suis* (worm) infection [51]. These data, combined with our findings, suggest an evolutionary role for IL11 in type 1 and 2 immune responses and epithelial barrier function. Thus, IL11 upregulation in inflammatory human diseases is unlikely to be a compensatory response, as suggested previously, and can instead be expected to contribute to allergy and inflammation as part of a broader alarmin response.

We conclude that IL11 activates two distinct signaling and biological pathways in fibroblasts: (1) an early STAT3-driven pro-inflammatory response that causes the secretion of neutrophil, monocyte and lymphocyte chemotaxins and primes the IL33 alarmin response and (2) a late ERK/LKB1/AMPK/mTOR-dependent pathway of fibrogenic protein translation and mesenchymal transition. The temporal dynamics of these pathways may fit with the response of the stroma to infection: initial pathogen killing (inflammation) followed later by pathogen sequestration (fibrosis).

Our data also show that IL11-stimulated fibroblasts can exhibit either pro-inflammatory or pro-fibrotic phenotypes depending on the duration of their exposure to IL11. Thus, the single-cell RNA-sequencing designation of separate fibroblast populations may in fact represent a single fibroblast type assayed across a continuum of potential phenotypes. We end by suggesting the IL11/IL33 axis as a potential therapeutic target for fibro-inflammatory diseases such as colitis, systemic sclerosis and rheumatoid arthritis.

## 4. Materials and Methods

### 4.1. Antibodies

Phospho-AKT (4060, Cell Signaling Technology (CST), Danvers, MA, USA), AKT (4691, CST, Danvers, MA, USA), phospho-ERK1/2 (4370, CST, Danvers, MA, USA), ERK1/2 (4695, CST, Danvers, MA, USA), GAPDH (2118, CST, Danvers, MA, USA), IL33 (ab207737, Abcam, Cambridge, UK), ⍺-SMA (19245, CST, Danvers, MA, USA), phospho-STAT3 (4113, CST, MA, Danvers, USA), STAT3 (4904, CST, Danvers, MA, USA), mouse HRP (7076, CST, Danvers, MA, USA), rabbit HRP (7074, CST, Danvers, MA, USA). All primary antibodies were used at 1:1000 dilution in TBST, and secondary antibodies were diluted 1:2000 in TBST containing 3% BSA.

### 4.2. Recombinant Proteins

Recombinant human IL11 (rhIL11, UniProtKB:P20809) was synthesized without the signal peptide using a mammalian expression system by Genscript (Piscataway, NJ, USA) with >95% purity by SDS-PAGE, effective dose 50 (ED_50_) of 5 ng/mL and bacterial endotoxin level of <0.2 EU/μg. Lyophilized rhIL11 was dissolved in PBS with 15 min of gentle agitation prior to use.

### 4.3. Chemicals

Stattic (S7947, Sigma-Aldrich, St. Louis, MO, USA), U0126 (9903, CST, Danvers, MA, USA).

### 4.4. Cell Culture

Cells were grown and maintained at 37 °C and 5% CO_2_. Primary human kidney fibroblasts (P10666, Lot 20115ty, InnoProt, Derio, Spain) isolated from a healthy human kidney (59-year-old male) were grown and maintained in fibroblast medium-PLUS (P60108-PLUS, Innoprot, Derio, Spain). Primary human lung fibroblasts (56-year-old male; CC-2512, Lonza, Basel, Switzerland) were grown and maintained in FGM-2 complete medium that contained fibroblast basal medium (CC-3131, Lonza, Basel, Switzerland) and the recommended growth supplements (FGM™-2 SingleQuots™, CC-4126, Lonza, Basel, Switzerland). Primary human skin fibroblasts (31-year-old female; 2320, ScienCell, Carlsbad, CA, USA) were grown in complete fibroblast medium consisting of basal medium (2301, ScienCell, Carlsbad, CA, USA), 2% fetal bovine serum (0010, ScienCell, Carlsbad, CA, USA), 1% fibroblast growth supplement (2352, ScienCell, Carlsbad, CA, USA), and 1% penicillin–streptomycin (0503, ScienCell, Carlsbad, CA, USA). All the experiments were carried out at P3. Cells were serum-starved for 16 h prior to stimulations with 10 ng/mL of IL11 for different durations and/or in the presence of inhibitors as outlined in the main text or figure legends.

### 4.5. RNA Sequencing

RNA-sequencing libraries were prepared and data were processed as previously described [52]. Raw sequencing data (.bcl files) were demultiplexed into fastq files with Illumina’s bcl2fastq v2.16.0.10 (Illumina, San Diego, CA, USA) based on unique index pairs. Adaptors and low-quality reads were trimmed using Trimmomatic V0.36 (USADELLAB, Aachen, Germany) [53], retaining reads >20 nucleotides. Read mapping was carried out using STAR v2.5.2b (Cold Spring Harbor Laboratory, Cold Spring Harbor, NY, USA) [54] to the Ensembl Human GRCh38 v86 reference genome using standard parameters for full-length RNA-seq in the ENCODE project. Read counting was carried out using featureCounts [55] to obtain gene-level quantification of genomic features: featureCounts -t exon -g gene_id -O -s 2 -J -p -R -G. Quality checks for sequencing and mapping data were carried out using FastQC v0.11.5 (Babraham Institute, Cambridge, UK) [56] and MultiQC (Ewels et.al., Stockholm, Sweden) [57]. Principal component analysis was carried out using the top 500 genes ranked by variance across samples. Differential expression (DE) was performed with DESeq2 v1.14.1 (https://bioconductor.org/packages/release/bioc/html/DESeq2.html) [58] using raw read counts from featureCounts. Baseline samples were used as the reference level for each paired two-group comparison with 1 h, 6 h and 24 h samples. Gene set enrichment analysis was carried out using fgsea R package, MSigDB hallmark and gene ontology sets with 100,000 iterations [59,60]. The “stat” column of the DESeq2 result output was used to rank the genes as inputs for the enrichment analysis.

### 4.6. Western Blot

Western blot was carried out on total protein extracts from HKFs, HLFs and HSFs. Cells were lysed in radioimmunoprecipitation assay (RIPA) buffer containing protease and phosphatase inhibitors (A32965 and A32957, Thermo Fisher Scientifics, San Francisco, CA, USA), followed by centrifugation to clear the lysate. Protein concentrations were determined by Bradford assay (Bio-Rad, Watford, UK). Equal amounts of protein lysates were separated by SDS-PAGE, transferred to a PVDF membrane and subjected to immunoblot analysis for various antibodies as outlined in the main text, figures or and/or figure captions. Proteins were visualized using the SuperSignal™ West Femto Maximum Sensitivity Substrate (Thermo Fisher Scientific, Waltham, CA, USA) with the appropriate secondary antibodies: anti-rabbit HRP or anti-mouse HRP.

### 4.7. Quantitative Polymerase Chain Reaction (qPCR)

Total RNA was extracted from cells using an RNeasy mini kit (74106, Qiagen, Germantown, MD, USA) for purification, and cDNAs were synthesized with an iScript^TM^ cDNA synthesis kit (1708841, Bio-Rad, Watford, UK) according to the manufacturer’s instructions. Gene expression analysis was performed on duplicate samples with the following primers: Hs04931857_m1 (*IL33*) and Hs02786624_g1 (*GAPDH*) using the TaqMan (Applied Biosystems, Waltham, MA, USA) technology on a StepOnePlus^TM^ (Applied Biosystem, Waltham, MA, USA) qPCR machine over 40 cycles. Expression data were normalized to *GAPDH* mRNA expression, and fold change was calculated using the 2^−∆∆Ct^ method.

### 4.8. Proteomic Analysis Using Olink Proximity Extension Assay

HKFs were seeded at a density of 2.5 × 10^5^ cells/well at 1 mL/well into 6-well plates. Following 0, 1, 6 or 24 h stimulation with IL11, the culture supernatants were collected, and randomized samples were sent to the Olink Proteomics (Uppsala, Sweden) for the commercially available proximity extension assay-based technology measurement [61]. Samples were analyzed in one batch using the 92-protein inflammation panel. Briefly, the binding of paired cDNA-tagged antibodies that were directed against the targeted proteins in the supernatant led to the hybridization of the corresponding DNA oligonucleotides allowing subsequent extension by a DNA polymerase. The protein levels were subsequently quantified using real-time qPCR. Protein concentrations expressed as normalized protein expression (NPX; log_2_ scale) were used for analysis. Proteins with concentrations below the limit of detection (calculated via a negative control serum-free medium) were excluded from analysis.

## Figures and Tables

**Figure 1 ijms-23-08900-f001:**
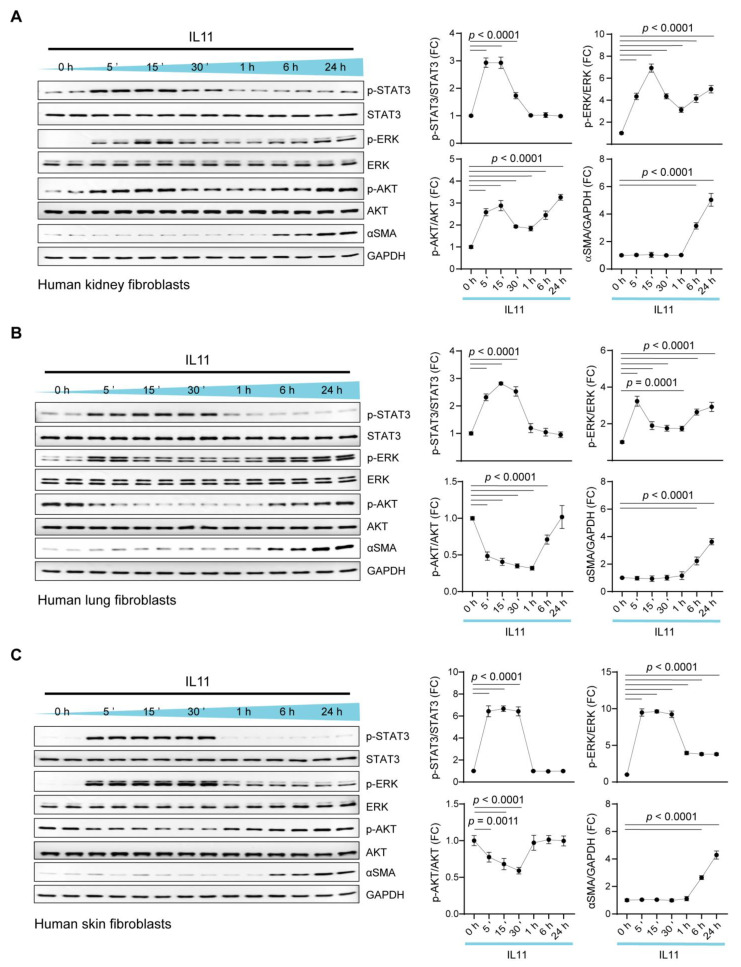
IL11 induces transient and early STAT3 phosphorylation but sustains ERK activation in fibroblasts from three different organs. Western blots (WB) and densitometry analyses of p-STAT3, STAT3, p-ERK, ERK, p-AKT, AKT, ɑSMA and GAPDH in IL11 (10 ng/mL)-stimulated primary human (**A**) kidney fibroblasts (HKFs), (**B**) lung fibroblasts (HLFs) and (**C**) skin fibroblasts (HSFs) across time points (5 m—24 h). Data are shown as mean ± SD; one-way ANOVA with Dunnett’s correction (n = 4 biological replicates), FC: Fold change.

**Figure 2 ijms-23-08900-f002:**
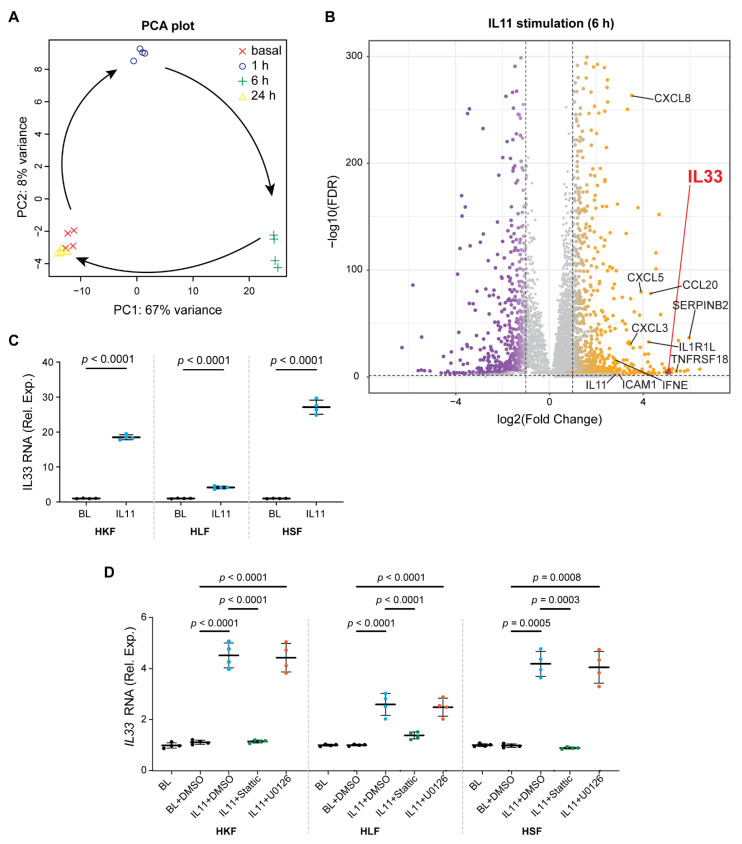
Fibroblasts stimulated with IL11 acquire a pro-inflammatory transcriptome and upregulate IL33 in a STAT3-dependent manner. (**A**,**B**) Data for RNA sequencing (RNA-seq) experiments on HKF stimulated with IL11 (10 ng/mL) for 0 (basal), 1, 6 and 24 h. (**A**) Principal component analysis (PCA) of RNA-seq across the time series. PC1 and PC2 account for 67% and 8% of the variance in the gene expression, respectively. (**B**) Volcano plots displaying fold change (FC) of genes between baseline and IL11-stimulated cells at 6 h time point. Dashed lines are drawn to define the restriction of log2 FC of 1 (vertical) and −log10 FDR of 0.05 (horizontal). Upregulated, downregulated and non-differentially expressed genes are labeled in orange, purple and gray, respectively. IL33 is annotated separately for clarity. (**C**) Relative mRNA expression (Rel. Exp.) of *IL33* in HKF, HLF and HSF following 6 h stimulation of IL11 compared with basal (BL) by qPCR; two-tailed Student’s *t*-test. (**D**) Relative *IL33* mRNA expression in BL and IL11-stimulated HKF, HLF and HSF in the presence of either DMSO, Stattic (STAT3 inhibitor, 2.5 µM) or U0126 (MEK inhibitor, 10 µM); one-way ANOVA with Tukey’s correction. (**C**,**D**) Data are shown as mean ± SD (n = 4 biological replicates).

**Figure 3 ijms-23-08900-f003:**
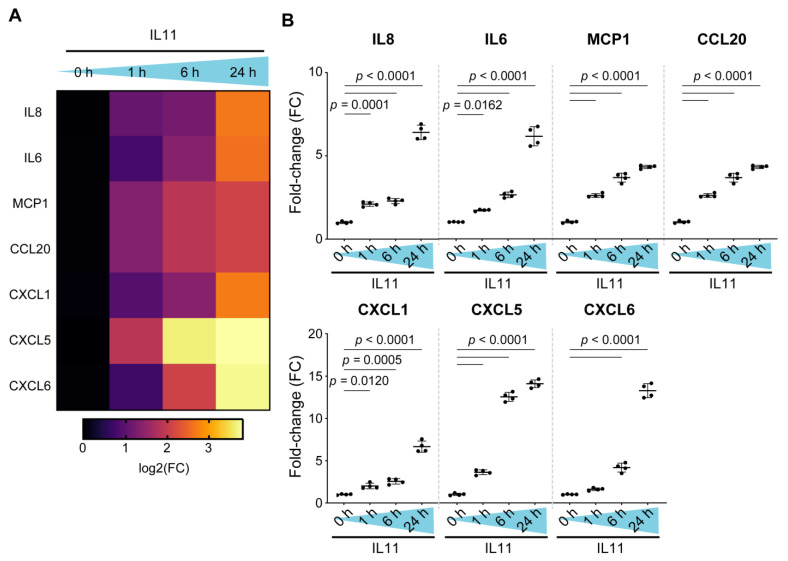
Human kidney fibroblasts stimulated with IL11 secrete a range of pro-inflammatory and chemotactic factors. (**A**) Pro-inflammatory cytokine and chemokine gene expression heatmap. (**B**) Relative levels of IL8, IL6, MCP1, CCL20, CXCL1, CXCL5 and CXCL6 in the supernatant of IL11-stimulated HKF across time points as measured by Olink proximity extension assay. Data are shown as mean ± SD (n = 4 biological replicates); one-way ANOVA with Dunnett’s correction, FC: Fold change.

**Figure 4 ijms-23-08900-f004:**
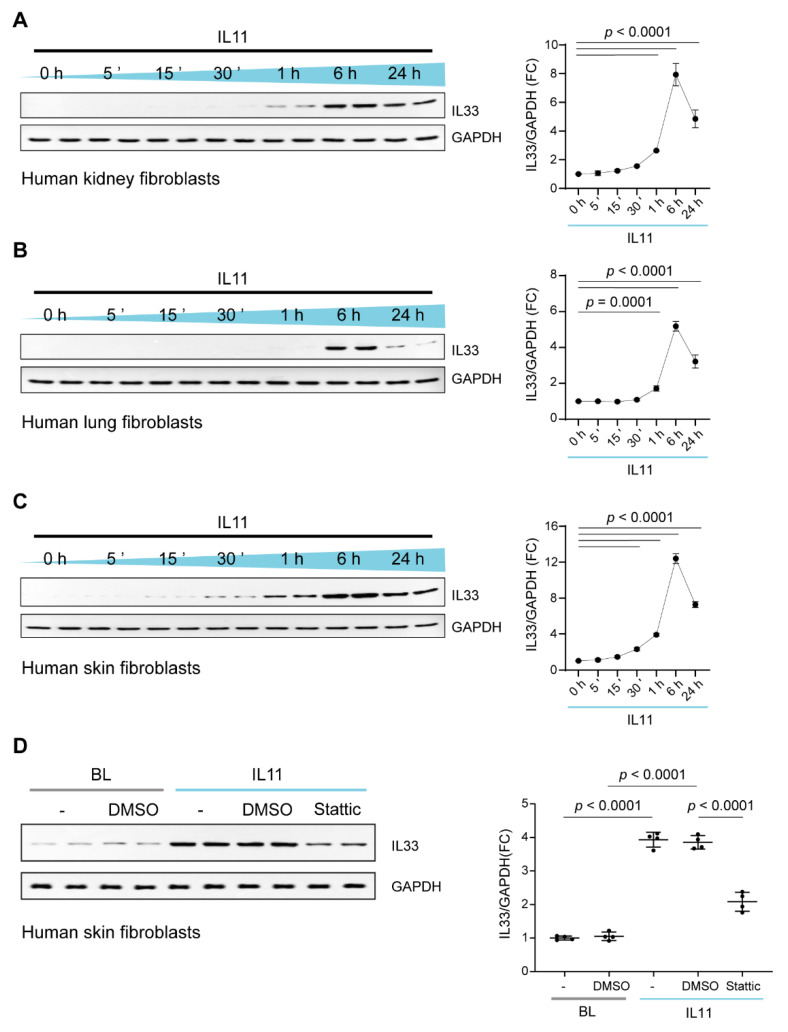
IL11 stimulates IL33 upregulation in kidney, skin, and lung fibroblasts. WB and densitometry analyses of IL33 relative to GAPDH expression in (**A**) HKFs, (**B**) HLFs and (**C**) HSFs following IL11 (10 ng/mL) stimulation for 5 m, 15 m, 30 m, 1 h, 6 h and 24 h and in (**D**) basal (BL) and IL11-stimulated HSFs in the presence of DMSO or Stattic (2.5 µM) or U0126 (MEK inhibitor, 10 µM) for 1 h; data are shown as mean ± SD (n = 4 biological replicates). (**A**–**C**) One-way ANOVA with Dunnett’s correction; (**D**) one-way ANOVA with Tukey’s correction, FC: Fold change.

## Data Availability

All data are provided in the manuscript and Appendix A. Uncropped blots are provided as Appendix A. Raw RNAseq data and gene-level counts will be uploaded onto the NCBI Gene Expression Omnibus database upon acceptance.

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
