# Peer review of "IL11 Stimulates IL33 Expression and Proinflammatory Fibroblast Activation across Tissues"

_ijms, 2022, doi:10.3390/ijms23168900_

Round 1

Reviewer 1 Report

Widjaja and colleagues present a research on the role of IL-11 in terms of anti-inflammatory activity in distinct cell culture experiments. The authors observed increased levels of antiinflammatory parameters and correlated these with proteomic datasets. The relation to IL-33 was highlighted which significantly influences further translational aspects and which should be considered in future aims. The manuscript is in summary well written, the introduction lacks on important aspects of the signaling cascade of IL-11 and in relation to IL-33, important reviews from Nature Immunology which were recently published should be cited. 

The aims whether human fibroblasts were used and not other cell types should be discussed in detail, as the authors would also expect same results in epithelial cell culture models?

The rational of proteomics should be written.

The discussion section is too short and does not include the interesting results which are correlated with the literature. Moreover, the methods section should precisely describe each method.

Author Response

Widjaja and colleagues present a research on the role of IL-11 in terms of anti-inflammatory activity in distinct cell culture experiments. The authors observed increased levels of antiinflammatory parameters and correlated these with proteomic datasets. The relation to IL-33 was highlighted which significantly influences further translational aspects and which should be considered in future aims. The manuscript is in summary well written, the introduction lacks on important aspects of the signaling cascade of IL-11 and in relation to IL-33, important reviews from Nature Immunology which were recently published should be cited.

Response: As requested by the Reviewer, we have now expanded the introduction to include recent and relevant reviews from Nature Immunology. We also mention more fully the relationship between IL11 and IL33, and its implications, in the discussion.

- The aims whether human fibroblasts were used and not other cell types should be discussed in detail, as the authors would also expect same results in epithelial cell culture models?

Response: In this study, we set out to define the role of IL11 in pro-inflammatory fibroblast activation or inhibition to address an outstanding dilemma in the literature. To ascribe robustness, we did these studies across three distinct fibroblast types from human lung, skin and kidney. This study, the first to identify a pro-inflammatory role of IL11 in a stromal cell type, does not extend to study immune cells, epithelial cells or any other cell type, which is beyond the aims and scope of the current manuscript.   

- The rational of proteomics should be written.

Response: We are unclear as to what the reviewer means by this. We choose to study the proteome as this has distinct properties to and may complement the transcriptome, as we found. If the reviewer refers instead to the methodology, this is described in the methods under the heading “Proteomic analysis using Olink proximity extension assay”. This is a proprietary assay from Olink, a technology used in over 900 published manuscripts. (https://www.olink.com/products-services/analysis-service/).

- The discussion section is too short and does not include the interesting results which are correlated with the literature. Moreover, the methods section should precisely describe each method.

Response: The discussion has been adjusted but we are keen to keep this a concise report and not to overinterpret this first description of IL11 on pro-inflammatory fibroblast activation. The methods have been expanded further.

Reviewer 2 Report

Sir, 

I have reviewed the manuscript " IL11 stimulates IL33 expression and proinflammatory fibroblast activation" submitted by Anissa A Widjaja on behalf of Prof Stuart Alexander Cook´s research group. The manuscript follows an immensely interesting topic of the IL-6 signalling family.  Indeed, IL-11 is something less appreciated member of this large family. Thus, it is worthy of attention as various aspects of its biology remain largely unknown or enigmatic. 

Generally, the article is scientifically sound and well written. However, I believe that there are also some possible improvements necessary. 

Relevant to the Introduction

a) the authors use the term "stroma". I see that the authors use it in the most classical way and relate it to the structure of any parenchymatous organ. For the sake of clarity, this should be explained.  Recently, many readers could be more familiar with the modern (and somewhat restricted) meaning related to the "cancer stroma" only. 

b)  the authors present fibroblasts of several types (renal, lung, dermal). However, it should also be briefly mentioned that fibroblasts are a largely diverse cell type in the human body. Fibroblasts can be of different embryonic origins (lateral plate mesoderm, cephalic mesoderm, neural crest), and this can largely influence their functionality. This can be visible in e.g. Figure 4 - skin fibroblasts vs. lung fibroblasts in IL33 production dynamics.  Moreover, I would also recommend considering donor age as an important factor. The authors can easily find published articles describing how skin fibroblasts differ in function in newborns and in senior patients. The authors use commercially available cells (at least one is fully described as male 59 y.o. - the rest is not known/mentioned). This might be another aspect for consideration. Importantly, the principal difference was observed in wound healing capacity and fibroblastic potential. 

c) The authors focus primarily on IL-11 and IL-33, but I find it very interesting that there is a stereotypical increase in a border panel, namely consisting of IL-6, IL-8 and CXCL1. It is a long-known complex fibroblast response observed in cancer and also, to some extent, in healing. This should be emphasised. 

d) It would be interesting to add to the discussion also a small comparison to the paradigmatic representative of this family - IL-6. It was described in detail that canonic and trans-signalling in IL-6 can result in pro- and also anti-inflammatory responses. It is context-dependent, and this factor must also be considered in the case of IL-11 signalling. 

e)  I am a bit puzzled by the paragraph in lines 72-81. This is dealing with an immensely interesting induction of myofibroblasts. IL-11 is  - broadly speaking - not considered to be the most relevant regulator now. Generally, it is the responsibility of TGFb-signalling. However, I do not find any attempt to link these two fibroblast regulators in the article. Moreover, the authors detect a very prominent increase in the course of 24h following stimulation. They also claim (quote): the expression of alpha-smooth muscle actin (É‘SMA), which occurs in the absence of detectable ACTA2 transcript increase in vitro [6,8]. At least in dermal fibroblasts, such prompt dynamics of fibroblast-to-myofibroblast transformation is very surprising,  or rather unlikely. This must be corroborated by a morphological study. Western blotting is a very tricky method for fibroblast assessment. You should see a well-organised cytoskeleton before you make a definite judgement. Pomylerised or unpolymerised - this is the question. Without proper cell morphology, it is advisable to stay silent about the actual percentage of myofibroblasts in the fibroblastic population. 

To conclude, the topic is interesting. The wet work seems to be well described and organised. There are some corrections necessary and -  maybe - some small additional experiments could increase the robustness of this manuscript. I am happy to see the revision at the nearest possible time. 

Author Response

Sir,

I have reviewed the manuscript " IL11 stimulates IL33 expression and proinflammatory fibroblast activation" submitted by Anissa A Widjaja on behalf of Prof Stuart Alexander Cook´s research group. The manuscript follows an immensely interesting topic of the IL-6 signalling family.  Indeed, IL-11 is something less appreciated member of this large family. Thus, it is worthy of attention as various aspects of its binoology remain largely unknown or enigmatic.

Generally, the article is scientifically sound and well written. However, I believe that there are also some possible improvements necessary.

Relevant to the Introduction

  1. a) the authors use the term "stroma". I see that the authors use it in the most classical way and relate it to the structure of any parenchymatous organ. For the sake of clarity, this should be explained. Recently, many readers could be more familiar with the modern (and somewhat restricted) meaning related to the "cancer stroma" only.

Response: The term “stroma” is now introduced only in the discussion and has been defined more clearly.

  1. b) the authors present fibroblasts of several types (renal, lung, dermal). However, it should also be briefly mentioned that fibroblasts are a largely diverse cell type in the human body. Fibroblasts can be of different embryonic origins (lateral plate mesoderm, cephalic mesoderm, neural crest), and this can largely influence their functionality. This can be visible in e.g. Figure 4 - skin fibroblasts vs. lung fibroblasts in IL33 production dynamics. Moreover, I would also recommend considering donor age as an important factor. The authors can easily find published articles describing how skin fibroblasts differ in function in newborns and in senior patients. The authors use commercially available cells (at least one is fully described as male 59 y.o. - the rest is not known/mentioned). This might be another aspect for consideration. Importantly, the principal difference was observed in wound healing capacity and fibroblastic potential.

Response: We agree that fibroblasts are diverse in embryonic origin and can vary in their response to stimulation depending on their embryonic origin, organ location, sex, age and disease context. For this very reason, we studied three separate types of fibroblasts (lung, kidney and skin) from different donors to determine if IL11 effects on fibroblast inflammation were unique to one fibroblast type or common to all. It is the case that factors with strong effect on fibroblast biology have conserved (but not identical) effects across fibroblasts, for instance TGFêžµ1 is pro-fibrotic in fibroblasts from all organs and across all species. We observed that IL11 has consistent effects on some signalling aspects (STAT3 and ERK) but not on others (Akt), which infers conserved effects of IL11 on STAT3/ERK activation (but not Akt) across fibroblast types. We also observed consistent, variable in magnitude and duration, effects of IL11-induced upregulation of IL33 at the RNA and protein levels that ascribes robustness to this response as well.  As per the Reviewers comment, we have added the details of age and sex for primary human lung and skin fibroblasts in the materials and methods section.

  1. c) The authors focus primarily on IL-11 and IL-33, but I find it very interesting that there is a stereotypical increase in a border panel, namely consisting of IL-6, IL-8 and CXCL1. It is a long-known complex fibroblast response observed in cancer and also, to some extent, in healing. This should be emphasised.

Response: We have added a sentence to discuss this aspect in the revised discussion.

  1. d) It would be interesting to add to the discussion also a small comparison to the paradigmatic representative of this family - IL-6. It was described in detail that canonic and trans-signalling in IL-6 can result in pro- and also anti-inflammatory responses. It is context-dependent, and this factor must also be considered in the case of IL-11 signalling.

Response: We, and others, are not convinced that there is a true biological role for IL11 trans-signalling, indeed our data to date suggests this does not play a role in physiology or disease (PMID: 33397952). Artificial trans-signalling constructs of IL11 have the same effect as IL11 cis signaling, in our hands. We therefore prefer not to discuss this controversy, in the current context.  

  1. e) I am a bit puzzled by the paragraph in lines 72-81. This is dealing with an immensely interesting induction of myofibroblasts. IL-11 is - broadly speaking - not considered to be the most relevant regulator now. Generally, it is the responsibility of TGFb-signalling. However, I do not find any attempt to link these two fibroblast regulators in the article. Moreover, the authors detect a very prominent increase in the course of 24h following stimulation. They also claim (quote): the expression of alpha-smooth muscle actin (É‘SMA), which occurs in the absence of detectable ACTA2 transcript increase in vitro [6,8]. At least in dermal fibroblasts, such prompt dynamics of fibroblast-to-myofibroblast transformation is very surprising,  or rather unlikely. This must be corroborated by a morphological study. Western blotting is a very tricky method for fibroblast assessment. You should see a well-organised cytoskeleton before you make a definite judgement. Pomylerised or unpolymerised - this is the question. Without proper cell morphology, it is advisable to stay silent about the actual percentage of myofibroblasts in the fibroblastic population.

Response: We find it hard to understand what the reviewer refers to in this comment and refer him to the extensive literature from our own lab and from other labs, which we cited, on the effects of IL11 on fibroblast biology in vitro and in vivo and the interaction between TGFêžµ1 and IL11, which is most clearly defined in the already published manuscripts.

To conclude, the topic is interesting. The wet work seems to be well described and organised. There are some corrections necessary and -  maybe - some small additional experiments could increase the robustness of this manuscript. I am happy to see the revision at the nearest possible time.

Reviewer 3 Report

IL11 is a stromal derived protein that has been implicated in a wide range of disease processes including fibrosis and cancer. The authors state the general understanding of IL11 function is as an anti-inflammatory factor. The purpose of this manuscript is to demonstrate that IL11 stimulation of primary human fibroblasts from kidney, dermis, and lung results in transcriptional increases in inflammatory genes including IL33 and IL1R1 using RNAseq and proteomic analyses. These data support that IL11 can induce pro-inflammatory signaling in fibroblasts.

Comments

Although the authors argue that the current consensus view of IL11 is anti-inflammatory, there are a considerable number of manuscripts in PUBMED that suggest IL11 is inflammatory. Perhaps the authors can take a different direction when describing the current view of IL11.

The investigators should provide details regarding the purification and validation of IL11. Is the preparation tested for endotoxin levels? 

What is the vehicle for IL11 and were vehicle treated controls used? 

More information should be provided regarding how fibroblast cells are generated and verified from each company?  The authors should also provide information regarding the expression of IL11 receptors on these primary fibroblasts. IHC or flow cytometry would provide the most meaningful information.

Antibody concentrations should be provided.

Although normalization for westerns is provided it is odd that instead of reblotting the same gel it appears that separate gels are used for the phospho and unphosphorylated proteins.  While the data appears rigorous, the authors should consider this in future data analyses.

Author Response

IL11 is a stromal derived protein that has been implicated in a wide range of disease processes including fibrosis and cancer. The authors state the general understanding of IL11 function is as an anti-inflammatory factor. The purpose of this manuscript is to demonstrate that IL11 stimulation of primary human fibroblasts from kidney, dermis, and lung results in transcriptional increases in inflammatory genes including IL33 and IL1R1 using RNAseq and proteomic analyses. These data support that IL11 can induce pro-inflammatory signaling in fibroblasts.

Comments

- Although the authors argue that the current consensus view of IL11 is anti-inflammatory, there are a considerable number of manuscripts in PUBMED that suggest IL11 is inflammatory. Perhaps the authors can take a different direction when describing the current view of IL11.

Response: The pervasive view, as summarised in the papers we cited, has been that IL11 is anti-inflammatory. For this reason it was used as an anti-inflammatory agent in a number of clinical trials. More recently, IL11 has been suggested to be pro-inflammatory in some contexts and we cite the publications supporting this view. In this way, we present IL11 in both its anti-inflammatory (historical) and pro-inflammatory (more recent) contexts in the introduction and the discussion.

- The investigators should provide details regarding the purification and validation of IL11. Is the preparation tested for endotoxin levels?

Response: These details have been added in the methods section. We highlight that the recombinant human IL11 that we used has specific IL11/IL11RA dependent activity as incubation of fibroblasts with this reagent in the presence of either anti-IL11 or anti-IL11RA antibodies prevents both STAT3 and ERK activation (PMID: 34651016). That is to say, the biological activity of this compound is entirely IL11 dependent and not related to any impurity that might be present.

-What is the vehicle for IL11 and were vehicle treated controls used?

Response: Lyophilized recombinant human IL11 (rhIL11) was dissolved in PBS to make a 10 µg/ml protein stock. For stimulation, 1 µl of the protein stock was added per ml of the respective cell culture media resulting in a final concentration of 10 ng/ml. Vehicle control for IL11 (PBS) was not used in this study as adding 1 µl of PBS/ml of buffered media has no effect on fibroblasts, or any other cell.

More information should be provided regarding how fibroblast cells are generated and verified from each company? The authors should also provide information regarding the expression of IL11 receptors on these primary fibroblasts. IHC or flow cytometry would provide the most meaningful information.

Response: We have provided additional information on the fibroblasts in the methods. All fibroblasts studied to date from all tissues express IL11RA, as described in Schafer et al, 2017 (PMID: 29160304). We also refer the reviewers to PMID: 34570432 and PMID: 33590875, for immunofluorescence data showing the IL11RA expression on the primary human lung and dermal fibroblasts, respectively that were used in this study (the same catalogue and lot number of both cell types were also used in those published study).

Antibody concentrations should be provided.

Response: These are now provided.

Although normalization for westerns is provided it is odd that instead of reblotting the same gel it appears that separate gels are used for the phospho and unphosphorylated proteins.  While the data appears rigorous, the authors should consider this in future data analyses.

Response: This is a standard approach that we have used over the last 20 years. We usually  reprobe the same gel for the second protein if it is of a different size from the first protein to avoid any signal mis-interpretation but this is a personal preference.

Reviewer 4 Report

In this manuscript Widjaja et al. have studied the actions of IL-11 on human proinflammatory fibroblasts from kidney, lung and skin. They found that IL-11 increased STAT-3 signalling and ERK activation, giving to a characteristic pattern of pro-inflammatory proteins, with marked upregulation of IL-33 and chemotactic factors for neutrophil, monocyte and lymphocyte migration. Moreover, they found that IL-33 expression by IL-11 was dependent on STAT-3 activation but not on ERK. The authors make evident the role that IL-11 plays in inflammation, underscoring part of the mechanisms by which it works.

Major comment

The study by Widjaja et al. represents an interesting and accurate approach to clarify whether IL-11 exerts pro-inflammatory or anti-inflammatory actions in human cells. For this purpose, their studies have been focussed on human fibroblasts as key players of tissue inflammatory responses. The manuscript has been written in a sober style yet it is reach of valuable information that may help to understand some of the pathophysiological consequences of IL-11-mediated STAT-3 activation.

Some specific points

- Fig. 1A: Which is the meaning of the increased activation of ERK after 1h IL-11 treatment in lung and kidney fibroblasts? Why does it not occur in skin fibroblasts?

The same for Akt in all kind of fibroblasts studied.

Regarding the expression of αSMA with the second wave of ERK activation induced by IL-11, results seem merely correlative. Alternate use of ERK and STAT-3 inhibitors seems required to unambiguously demonstrate the authors’ hypothesis that IL-11-induced αSMA expression in MT takes place by the second wave of IL-11-mediated ERK activation.

- Fig. 2B: Despite IL-33 is one of the most upregulated inflammatory genes from RNAseq analysis, volcano plot shows a very high FDR for it (almost no significant). Which was the rational to select this cytokine for ulterior analyses?

- Fig. 2C vs Fig. 4B: An apparent inconsistency is observed between IL-33 production at RNA and protein level in HLF stimulated with IL-11 for 6 h. Compared with other fibroblasts, why is this relative expression of IL-33 RNA notably lower than IL-33 protein? The authors should convincingly explain this issue.  

Author Response

In this manuscript Widjaja et al. have studied the actions of IL-11 on human proinflammatory fibroblasts from kidney, lung and skin. They found that IL-11 increased STAT-3 signalling and ERK activation, giving to a characteristic pattern of pro-inflammatory proteins, with marked upregulation of IL-33 and chemotactic factors for neutrophil, monocyte and lymphocyte migration. Moreover, they found that IL-33 expression by IL-11 was dependent on STAT-3 activation but not on ERK. The authors make evident the role that IL-11 plays in inflammation, underscoring part of the mechanisms by which it works.

Major comment

The study by Widjaja et al. represents an interesting and accurate approach to clarify whether IL-11 exerts pro-inflammatory or anti-inflammatory actions in human cells. For this purpose, their studies have been focussed on human fibroblasts as key players of tissue inflammatory responses. The manuscript has been written in a sober style yet it is reach of valuable information that may help to understand some of the pathophysiological consequences of IL-11-mediated STAT-3 activation.

Some specific points

- Fig. 1A: Which is the meaning of the increased activation of ERK after 1h IL-11 treatment in lung and kidney fibroblasts? Why does it not occur in skin fibroblasts?

The same for Akt in all kind of fibroblasts studied.

Response: Fibroblasts are diverse in embryonic origin and can vary in their response to stimulation depending on their organ location, sex, age and disease/physiological context. For this very reason we studied three separate types of fibroblasts (lung, kidney and skin) from separate donors to determine if IL11 effects on fibroblast inflammation were unique to one cell type or common to all. It is the case that factors with strong effect on fibroblast biology have conserved (but not identical) effects across fibroblasts, for instance TGFêžµ1 is pro-fibrotic in fibroblasts from all organs and across all species. We observed that IL11 has consistent effects on some signalling aspects (STAT3, ERK) but not on others (Akt), which infers conserved effects of IL11 STAT3/ERK (but not Akt) activation across fibroblast types. This led us to dissect STAT3 from ERK effects and to the discovery of STAT3-dependent IL33 upregulation by IL11.

Regarding the expression of αSMA with the second wave of ERK activation induced by IL-11, results seem merely correlative. Alternate use of ERK and STAT-3 inhibitors seems required to unambiguously demonstrate the authors’ hypothesis that IL-11-induced αSMA expression in MT takes place by the second wave of IL-11-mediated ERK activation.

Response: The mechanistic relationship between ERK activation and αSMA has been established in a large number of publications over the last five years. We cited these publications and discussed the context. We refer the reviewer to the citations and for yet further clarity have added some more discussion on this.  

- Fig. 2B: Despite IL-33 is one of the most upregulated inflammatory genes from RNAseq analysis, volcano plot shows a very high FDR for it (almost no significant). Which was the rational to select this cytokine for ulterior analyses?

Response: By RNA-seq analysis L33 was the most upregulated signalling factor (38-fold, P=9.8x10-5). We validated IL33 expression by qPCR across three separate fibroblast types at 1h and 6h after IL11 stimulation (P<0.0005 for all). Its upregulation was confirmed by Western blotting and its STAT3-dependency validated at both the RNA and protein level. 

- Fig. 2C vs Fig. 4B: An apparent inconsistency is observed between IL-33 production at RNA and protein level in HLF stimulated with IL-11 for 6 h. Compared with other fibroblasts, why is this relative expression of IL-33 RNA notably lower than IL-33 protein? The authors should convincingly explain this issue. 

Response: We refer the reviewer to the first point above. Fibroblasts from across tissues, sexes and donors are non-identical and we specifically use this diversity to determine if IL11 effects on inflammatory fibroblasts activation are conserved or not. We do not expect IL11 effects in different fibroblasts to be identical, indeed this is highly unlikely. What we show is a common effect of IL11 on inflammatory phenotypes (STAT3 signalling; IL33 upregulation) that is variable in magnitude and duration and differs at the RNA and protein levels, which is as expected and why we assess both RNA and protein expression in this study.

Round 2

Reviewer 1 Report

The authors have addressed my concerns and the impact of the originality has been significantly increased.